# Uniform Manifold Approximation and Projection for Clustering Taxa through Vocalizations in a Neotropical Passerine (Rough-Legged Tyrannulet, *Phyllomyias burmeisteri*)

**DOI:** 10.3390/ani10081406

**Published:** 2020-08-12

**Authors:** Ronald M. Parra-Hernández, Jorge I. Posada-Quintero, Orlando Acevedo-Charry, Hugo F. Posada-Quintero

**Affiliations:** 1Institución Educativa Normal Superior Fabio Lozano Torrijos, Falan, Tolima 732001, Colombia; 2Asociación Tolimense de Ornitología, Ibagué, Tolima 730005, Colombia; 3Institución Educativa Técnica María Auxiliadora, Fresno, Tolima 731560, Colombia; jorge.posada@sedtolima.edu.co; 4Facultad de Ciencias de la Educación y la Comunicación, Universidad Autónoma de Asunción, Asunción 001013, Paraguay; 5Colección de Sonidos Ambientales, Instituto de Investigación de Recursos Biológicos Alexander von Humboldt, Claustro de San Agustín, Villa de Leyva, Boyacá 154001, Colombia; oacevedo@humboldt.org.co; 6Asociación Colombiana de Ornitología, Bogotá, DC 111311, Colombia; 7Department of Biomedical Engineering, University of Connecticut, Storrs, CT 06269, USA

**Keywords:** *acrochordopus*, bioacoustics, dimensionality reduction, sound, taxonomic complex, tyrannidae, tyranninae, *tyranniscus*, voices, white-fronted tyrannulet

## Abstract

**Simple Summary:**

Recognizing the different species can help us better understand the nature. One way to differentiate bird species is the bird song. There are mathematical techniques that extract information from the bird songs, potentially allowing automatic differentiation of species. However, there is still a lack of techniques that use the extracted information and accurately differentiate individuals of different species. For the first time, we have used a technique called Uniform Manifold Approximation and Projection (UMAP) to identify the two taxonomic groups of a bird named Rough-legged Tyrannulet *Phyllomyias burmeisteri*, which is a species that can be found all the way from Costa Rica to Argentina. Although there is evidence of the existence of two taxonomic groups, previous studies have shown them to be difficult to distinguish. We collected Rough-legged Tyrannulet bird songs of 101 birds from 11 countries. UMAP allowed us to make a transformation of the multiple measures obtained from the bird song of each bird into just two values. Plotting the UMAP values of each bird in a two-dimensional graph, it turns out that UMAP was able to clearly identify the two taxonomic groups, which has been named as Rough-legged Tyrannulet *Phyllomyias burmeisteri* and White-fronted Tyrannulet *Phyllomyias zeledoni*. UMAP can potentially help the identification of other species difficult to classify.

**Abstract:**

Vocalizations from birds are a fruitful source of information for the classification of species. However, currently used analyses are ineffective to determine the taxonomic status of some groups. To provide a clearer grouping of taxa for such bird species from the analysis of vocalizations, more sensitive techniques are required. In this study, we have evaluated the sensitivity of the Uniform Manifold Approximation and Projection (UMAP) technique for grouping the vocalizations of individuals of the Rough-legged Tyrannulet *Phyllomyias burmeisteri* complex. Although the existence of two taxonomic groups has been suggested by some studies, the species has presented taxonomic difficulties in classification in previous studies. UMAP exhibited a clearer separation of groups than previously used dimensionality-reduction techniques (i.e., principal component analysis), as it was able to effectively identify the two taxa groups. The results achieved with UMAP in this study suggest that the technique can be useful in the analysis of species with complex in taxonomy through vocalizations data as a complementary tool including behavioral traits such as acoustic communication.

## 1. Introduction

Differentiating species is particularly challenging for taxa of birds in families that exhibit similar plumages and vocalizations, such is the case of the wrens (Troglodytidae) [1,2], tapaculos (Rhinocryptidae) [3,4,5], antbirds (Thamnophilidae) [6] and New World flycatchers (Tyrannidae) [7]. An interesting case is the Rough-legged Tyrannulet (*Phyllomyias burmeisteri*). Although the existence of two groups has been suggested [8,9], the species has presented taxonomic difficulties in classification [10]. Recently, principal component analysis (PCA) was used to group the subspecies of Rough-legged Tyrannulet complex from vocalizations [10]. The results further support the existence of two main groups; however, the study was not conclusive since an overlap between the groups existed in the PCA representation. A more sensitive analysis technique is needed to clearly separate both groups. 

Several statistical techniques have been proposed to effectively differentiate specimens from vocalizations. Such techniques aim to provide an improved classification of bird species with inconclusive scientific evidence to determine the taxonomic status of the species that are difficult to identify [11]. With the advent of machine learning and big data analysis, novel dimensionality reduction techniques have become powerful tools for data analysis enabling a better visualization and understanding of large, high dimensional datasets [12,13]. Dimensionality reduction plays an important role in multiple-dimension data analysis, as it is a fundamental technique for visualization and data processing. Deep mathematical frameworks and engineering tools, such as dimensionality reduction, have improved the way biological studies approach diversity variation [14,15,16]. For example, many techniques are available for biological applications looking into group differentiation through vocalizations. Statistical tools include testing the difference in standardized note measures by contrast (*t*-Test) [6], analysis of variance (ANOVA) [17], or multivariate ANOVA (MANOVA) [18]. Other statistical tools enable multidimensional analysis, such as PCA [19,20,21], discriminant analysis [3], among others [22]. 

Compared to other statistical tools, one of the advantages of dimensionality reduction is to allow the processing of massive data and to work a high number of dimensions, like the multiple bioacoustics measures obtainable from vocalizations. The development of novel and sensitive techniques for dimensionality reductions of quantitative variables obtained from acoustic signals would help to cluster taxa through vocalizations. Here, we have tested one of the most novel techniques for dimensionality reduction, the Uniform Manifold Approximation and Projection (UMAP), for detecting groups of subspecies within Rough-legged Tyrannulet.

There are several dimensionality reduction algorithms besides UMAP. The algorithms of dimensionality reduction can be divided into two categories [23]: those that preserve the distance structure within the data such as the PCA [24], multidimensional scaling [25] and Sammon mapping, and those that favor the preservation of local distances over the global distance like t-distributed stochastic neighbor embedding (t-SNE) [26], Isomap, LargeVis, Laplacia eigenmaps and diffusion maps [27]. Among the tools in the second group, the UMAP is a novel dimensionality reduction and visualization technique developed to preserve more stable and consistent structures than other methods and has shown superior performance compared to other tools (including t-SNE) [23,28,29]. UMAP constructs a high dimensional graph representation of the data and then optimizes a low-dimensional graph to be as structurally similar as possible [30]. 

We have aimed to evaluate the feasibility of UMAP for grouping subspecies from the analysis of vocalizations. To test the sensitivity of the technique, we have run UMAP analysis of a diverse (from Costa Rica to Argentina) set of samples of a Passerine bird vocalization. Our species model includes a complex taxonomy [11], but vocalizations could support its taxonomic treatments [10]. Thus, this article evaluates the suitability of UMAP for the separation of population groups, allowing a more meaningful visualization of the data, which can lead to a more conclusive separation of taxa. 

## 2. Materials and Methods

### 2.1. Species Model

Rough-legged Tyrannulet (*Phyllomyias burmeisteri* [*sensu lato*]) comprises a complex group of subspecies that are distributed in 6 zones of Central and South America (Figure 1): (1) *zeledoni*; Costa Rica (CR) and Panama (PA); (2) *wetmorei*; Sierra Nevada de Santa Marta and Sierra de Perijá, Colombia (CO)-Venezuela (VE) border; (3) *viridiceps*; northern mountains of VE; (4) *bunites*; the Pantepuis of VE; (5) *leucogonys*; along the Andes from CO to the south up to southeast of Peru (PE), through Ecuador (EC); (6) *burmeisteri*; with two isolated population in central Andean rainforest or Yungas at eastern Andes of Bolivia (BO) and northern Argentina (AR), and the Atlantic Forest in southeast and south of Brazil (BR), southeast of Paraguay (PY), Uruguay (UY), and the extreme northeast of Argentina (AR) [31]. 

This species has presented taxonomic difficulties in classification [10]. Rough-legged Tyrannulet sometimes is treated as two different species: White-fronted Tyrannulet *P. zeledoni* (*sensu lato*) (joining subspecies 1 to 5, see above) and *P. burmeisteri* (*sensu strictu*) (subspecies 6, above). Such separation has been somehow supported by preliminary acoustic analysis, White-fronted with longer and sharper notes, and Rough-legged with short and lower frequency notes [8,9]. Recently, Parra and Arias [10] used PCA to group the subspecies of this complex from vocalization. The results supported the existence of two main groups; however, the study was not conclusive since there was an overlap between the groups in the PCA representation, and the number of samples included only 42 individuals. A more sensitive analysis technique with larger sampled dataset is needed to clearly separate potential groups, allowing definitive conclusions.

### 2.2. Dataset

We analyzed the vocalizations of *N* = 101 individuals of Rough-legged Tyrannulet (*sensu lato*), whose localization ranged from Costa Rica to Uruguay (Figure 1, Appendix A). Our dataset corresponded to four of the six known taxa of the complex. These data were taken from the virtual repositories xeno-canto [32], Macaulay Library [33], Ecoregistros [34], and the Avian Vocalizations Center of Michigan State University [35]. From the 227 vocalizations available in those repositories (Appendix A), we defined three selection criteria for the inclusion of vocalization in our analysis. First, we used only vocalizations that include at least 6 notes. Second, we excluded those vocalizations with low quality regarding signal-to-noise ratio (lower differentiation with background noise). Third, we chose vocalizations from different geographic areas of each country, depending on the availability of vocalizations. Based on these criteria, we analyzed 100% of the data available from Bolivia, Panama, Uruguay, and Venezuela, and more than 75% of the vocalizations available from Colombia, Costa Rica, Paraguay, and Peru. Regarding the three countries with the highest number of vocalizations available, we included more than 30% of the vocalizations from Argentina, Ecuador, and Brazil (Appendix A). Appendix A includes location, taxa, elevation, and subject ID (within the specific catalog) of the audio samples of Rough-legged Tyrannulet (*sensu lato*) included in our analysis (Figure 1). For each vocalization of a given individual, we extracted a maximum of ten notes. Most vocalization had 10 or more notes available. For illustration purposes, we have provided two representative recordings in the Appendix A. For each note, we computed four acoustic parameters: (1) dominant frequency (*F-Dom*) in kHz; (2) note duration in seconds (*Delta-t*); (3) maximum frequency (*F-Max*); and (4) minimum frequency (*F-Min*). To extract these parameters, we used Raven version 1.2.1 [36]. Each individual corresponds to a row of the matrix *D*, as shown below. Notice that as we computed four measures for each of the ten notes, the matrix has forty columns (these are the dimensions to begin the UMAP analysis). The size of *D* is 101 (individuals) × 40 (dimensions). UMAP analysis was conducted in MATLAB (MathWorks, Inc., Natick, USA) [37], but there is a UMAP implementation available in R.
$$D=(\begin{array}{ccc} \begin{array}{c} \begin{array}{ccc} \begin{array}{cc} F-Dom_{1,1} & Delta-t_{1,1} \end{array} & \begin{array}{cc} F-Max_{1,1} & F-Min_{1,1} \end{array} & \begin{array}{cc} F-Dom_{2,1} & \begin{array}{cc} Delta-t_{2,1} & F-Max_{3,1} \end{array} \end{array} \end{array}\\ \begin{array}{ccc} \begin{array}{cc} F-Dom_{1,2} & Delta-t_{1,2} \end{array} & \begin{array}{cc} F-Max_{1,2} & F-Min_{1,2} \end{array} & \begin{array}{cc} F-Dom_{2,2} & \begin{array}{cc} Delta-t_{2,2} & F-Max_{3,2} \end{array} \end{array} \end{array} \end{array} & \cdots & \begin{array}{c} F-Min_{10,1}\\ F-Min_{10,2} \end{array}\\ \vdots & \ddots & \vdots \\ \begin{array}{ccc} \begin{array}{cc} F-Dom_{1,N} & Delta-t_{1,N} \end{array} & \begin{array}{cc} F-Max_{1,N} & F-Min_{1,N} \end{array} & \begin{array}{cc} F-Dom_{2,N} & \begin{array}{cc} Delta-t_{2,N} & F-Max_{3,N} \end{array} \end{array} \end{array} & \cdots & F-Min_{10,N} \end{array})$$
where the first sub-index represents the note, and the second sub-index represents the individual.

### 2.3. The Uniform Manifold Approximation and Projection (UMAP)

UMAP constructs a high dimensional graph representation of the data then optimizes a low-dimensional graph to be as structurally similar as possible [23]. In summary, UMAP constructs a weighted graph from the high dimensional data, with strength representing how connected a given point is to another, then projects this graph down to a lower dimensionality. UMAP algorithm is founded on three assumptions about the data: (1) the data is uniformly distributed on Riemannian manifold; (2) the Riemannian metric is locally constant or can be approximated as such; and (3) the manifold is locally connected. While the mathematics UMAP uses to construct the high-dimensional graph is advanced, the intuition behind them is remarkably simple. 

Here we are providing the basic details and simple exemplification of the functioning of UMAP. For a complete description of the UMAP algorithm, please see [23]. UMAP builds a representation of a weighted graph (the ‘fuzzy simplicial complex’), with edge weights representing the likelihood of connection of each pair of data points. To assess connectedness, UMAP extends a radius outwards from each point, connecting points when those radii overlap. Choosing this radius is critical because a radius too small will lead to small, isolated clusters, while too large radius will connect everything. To overcome this challenge, UMAP chooses a radius locally, based on the distance to each point’s ‘nth’ nearest neighbor. Figure 2 exemplifies the process. In this exemplifying two-dimension dataset, we have extended the radius at 22% (center) and 100% (right) of the distance of each point to its 5th neighbor. UMAP then makes the graph ‘blurred’ by decreasing the likelihood of connection as the radius grows. In Figure 2, notice that past the intersection with the first neighbor, the radius begins to get blurred, meaning that subsequent connections have less weight. Finally, by stipulating that each point must be connected to at least its closest neighbor, UMAP ensures that local structure is preserved in balance with the global structure. From the resulting graph, UMAP optimizes the layout of a low-dimensional analog to be as similar as possible. Furthermore, we performed k-medoids clustering [38] to test which cluster each individual belongs to. We used the squared Euclidean distance metric and the k-means++ algorithm for choosing initial cluster medoid positions [39]. Then, we map each record and compare the elevation between the groups with a Mann–Whitney test (given the non-normality of the data, as evidenced by the Shapiro test).

## 3. Results

We have found that UMAP clearly split the individuals into two groups during the dimensionality reduction from 40 to two dimensions in our dataset of Rough-legged Tyrannulet vocalizations (Figure 3). The values in the axis of the representation do not have a direct interpretation but represent the most similar projection of the original set of 40 dimensions into two dimensions. The contours of the two clusters showed a clear separation between them, which is congruent with the differential distribution between groups. The results suggest that the data met the assumptions of the UMAP algorithm. Furthermore, the results of the k-medoids clustering analysis, including the country of localization of each individual sample, showed segregation in the horizontal axis for the two-dimensional plot (Figure 3). The results support the separation of the two groups by geographical location (country). The first group (Group A) included individuals from Uruguay, Paraguay, Bolivia, Argentina, and Brazil. The second group (Group B) included individuals from Ecuador, Bolivia, Peru, Costa Rica, Colombia, Panama, and Venezuela.

For comparison purposes, we implemented PCA and t-SNE for dimensionality reduction and clustering (Figure 4). For PCA, the two first components are used as X- (first principal component, PC 1), and Y-axis (second principal component, PC 2). It can be observed that PCA provides some separation with a certain spread of the groups. Two PE individuals were included in Group A, which is very unlikely to be correct. As for t-SNE, one CR individual is clustered with Group A and one AR was included in Group B, as the separation between groups is not as clear as with UMAP.

When mapping our UMAP clustering results in the distribution of the species (Figure 5), there is a congruence in taxonomic treatment in two main groups: Group A, Rough-legged (*P. burmeisteri*), and Group B, White-fronted Tyrannulet (*P. zeledoni*). Group A included burmeisteri, while Group B included the subspecies *zeledoni*, *viridiceps*, and *leucogonys*. The resulting visualization allows direct analysis and inferring the distribution patterns of the individuals, considering the spatial locations of each individual. Of particular interest, the northern Bolivia distribution records seem to have almost sympatric populations for both groups, segregating individuals of *leucogonys* and *burmeisteri* subspecies in a multivariate space. These individuals of Rough-legged (Group A) could inhabit the dry and low areas with respect to those of White-fronted (Group B).

The two groups also differed in recorded elevation (Figure 6; Mann–Whitney test, *p* < 0.001). Records of White-fronted Tyrannulet were at a higher elevation, with a median 1198.5 m and interquartile range (IQR) from 1064.6 to 2166.3 m. Records of Rough-legged Tyrannulet (*sensu strictu*) were lower in elevation, with a median 665.2 m and IQR from 225.4 to 982.3 m. 

## 4. Discussion

For the first time, we have conducted a UMAP dimensionality reduction analysis to separate bird taxa based on acoustic signals. UMAP was more effective and sensitive than previously used techniques [10], as it allowed clearer grouping subspecies from taxa. The analysis framework based on UMAP showed more clearly than ever that the Rough-legged Tyrannulet complex could be effectively separated into two groups by their vocalizations, but our results require other analyses to have a better understanding of the taxonomy and evolution history within the group [41,42,43]. The accuracy of UMAP in this study suggests that the mathematical power of the technique can be useful in the analysis of vocalizations in species with complex taxonomy as a complementary tool including behavioral traits as acoustic communication.

As for the results of the two group identification using UMAP, the first group corresponded to individuals from southern South America (*P. burmeisteri*), distributed geographically in southeastern Bolivia, southern Brazil, southeastern Paraguay, Uruguay, the northeast of Argentina, and the foothills of the Andes from northwest Bolivia to northwest Argentina. The second group is distributed towards the north of South America and Central America (*P. zeledoni*) so that the limit is established where both species break into the north of Bolivia. Such treatment of these two groups has been used in several taxonomic arrangements, but there is still no consensus in the South American Classification Committee [11]. For example, the Andean group (*leucogonys*) has been proposed as different from Central America (*zeledoni*), but our UMAP analysis suggested these two groups to be combined by vocal traits. Treatment as species for these groups has not been supported based on a lack of published analysis of vocal differences [10,31]. We provide here a sensitive and robust analysis that separates acoustically two taxa within the complex. Further analysis must include individuals of subspecies not included here (*viridiceps*, *wetmorei*, and *bunites*), some with shared distribution of recently separated species complex of the humid Andes [42,43].

Our UMAP analysis included the Central America subspecies, *zeledoni*, the Venezuelan mountain, *viridiceps*, and Andean population, *leucogonys*, as a unique group distributed at a higher elevation (White-fronted Tyrannulet). Central America records ranged from 1182 to 3365 m, Venezuelan records from ~800 to 1150, and Andean records from 865 to 2000 m. Simultaneously, our UMAP analysis included the two allopatric populations of the Yungas and the Atlantic Forest as a differentiated group at a much lower elevation (Rough-legged Tyrannulet). The Yungas foothills records ranged from 100 to 2000 m, while the Atlantic Forest records ranged from 15 to 2300 m. In the Yungas the warm and humid air of the tropical humid forest is trapped and the level of precipitation is much higher than in surrounding areas [44]. At the northern but lower portion, the predominance of dry and semideciduous forests is relevant [45]. 

The central Andes or Yungas (Bosque Tucumán, Bolivia) and the Atlantic Forest are separated by a distance of 700 km, which exhibits an open vegetation corridor made up of Caatinga, the Cerrado and the Chaqueña regions [46,47,48,49]. However, these areas have high convergent characteristics and taxonomic similarities [46,47,50,51], which indicates that there was an ancient historical and biotic connection between these biomes [48,52,53], likely to concur during the Pleistocene [48,51,54]. Thus, the Yungas and the Atlantic Forest are considered to have functioned as a biodiversity refuge during the Pleistocene [55,56,57]. It has been postulated that the emerging species would be the product of the isolations originated by the climatic changes during the Quaternary, effective to allow speciation [58], the product of processes that include the elevation of the Andes. Such climatic fluctuations have resulted in habitat fragmentation during the Quaternary and the emergence of river and landscape barriers [59,60,61,62,63].

Similarly, it has been pointed out that the uprising of the Andes was an important driver of avian diversification because it gave rise to novel environments in which species could spread and diversify, where greater opportunities for allopatric differentiation in mountain areas were promoted [63,64,65]. Thus, some of these patterns contributed to the acoustic differentiation between Rough-legged and White-fronted tyrannulets, as has been suggested in some studies where these biomes have been considered as important promoters of species diversification [48,51,54,66,67].

Our work consolidates the postulates of two acoustic divergent groups [9,10], as well as recent classifications [8], but further genetic analysis must be included to confirm our conclusions. Furthermore, we have shown that UMAP can be used for performing such analyzes effectively. We expected that UMAP could help to classify other bird species or groups of organisms from the analysis of their acoustic traits. For instance, species that due to their morphological characteristics as plumage are difficult to separate taxonomically. However, it is important to apply and define this technique in organisms with more complex vocalizations, taking into account that animals emit acoustic signals that vary in complexity, which represents from a single repeated call (note), as in the case of our species model, to hundreds of different vocal elements [68].

Animal vocalizations have been previously projected onto a linear feature of spaces using principal component analysis [10,69]. In that scenario, as our results suggested, each successive dimension represents an orthogonal transformation capturing the maximum possible variance between the data [68,70]. Vocalizations can also be decomposed into characteristics using linear discriminant analysis [5] where characteristics are determined by their ability to explain variance in a specific dimension of data, such as individual identity [68,71]. However, the dimensionality reduction may also be non-linear, representing a greater number of data relationships (for example, the similarity between notes of birdsong). The usefulness of nonlinear dimensionality reduction techniques is just beginning to be studied in animal communication [68]. For example, using stochastic neighborhood inlay distributed in t (t-SNE) [26] as is the case study of the development description of the Zebra Finch (*Taeniopygia guttata*) [68,72]. Other studies have validated UMAP as a suitable dimensionality reduction technique, that plays an important role in data science, as the most effective visualization and analysis of multiple data to date [13,23], so that the statistical relationships generate more appropriate forms for the explanation of groups than some traditional linear techniques. 

In this study, we compared UMAP to other dimensionality reduction techniques, namely PCA and t-SNE. UMAP has provided better results as the separation of the two groups was clearer using UMAP, whereas PCA and t-SNE showed some overlap between groups (Figure 4). Furthermore, PCA and t-SNE assigned individuals to groups to which they are very unlikely to belong to. These results evidenced the unique advantages of UMAP, compared to PCA and t-SNE.

As for the limitations of the study, the dataset consisting of 101 individuals could be considered low to make conclusive claims. We have used a significant sample from the available recordings of the Rough-legged Tyrannulet *Phyllomyias burmeisteri* complex, as evidenced in the Appendix A. Nevertheless, UMAP was able to perform a dimensionality reduction that allowed us to separate the two groups more accurately than with PCA and t-SNE, which evidences the appropriateness of the technique.

## 5. Conclusions

We proposed a novel approach for acoustic analysis of subgroups using dimensionality reduction capacities of UMAP. UMAP allows the grouping of species taxa in a clear way, based on graphic analysis from acoustic data. The UMAP optimizes the low-dimensional analog graphic representation of the data, whose visualization is as similar as possible to the original structure of the data. The proposed UMAP analysis successfully defined the separation of Rough-legged Tyrannulet complex into two acoustic groups. Taxonomical and evolutionary outputs of our analysis could be supported with further work in phylogenetic approaches within the group.

## Figures and Tables

**Figure 1 animals-10-01406-f001:**
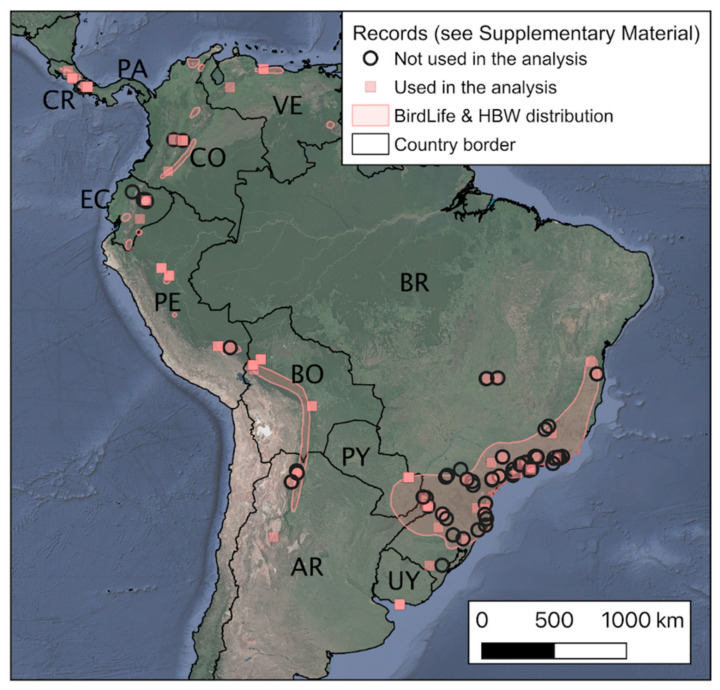
Distribution of Rough-legged Tyrannulet (*sensu lato*). Records used in the bioacoustics analysis with Uniform Manifold Approximation and Projection (UMAP) are represented by a pink square (Appendix A). Hollow circles are other acoustic records, not used here in the analysis. Country codes alpha-2 represent the countries with records.

**Figure 2 animals-10-01406-f002:**
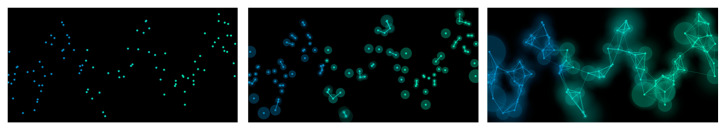
Illustration of UMAP process. Original data points (**left**), 22% of the distance to 5th neighbor (**center**), and 100% of the distance to 5th neighbor (**right**). Beyond the intersection with the first neighbor, the radius begins to get blurred, assigning less weight to subsequent connections [40].

**Figure 3 animals-10-01406-f003:**
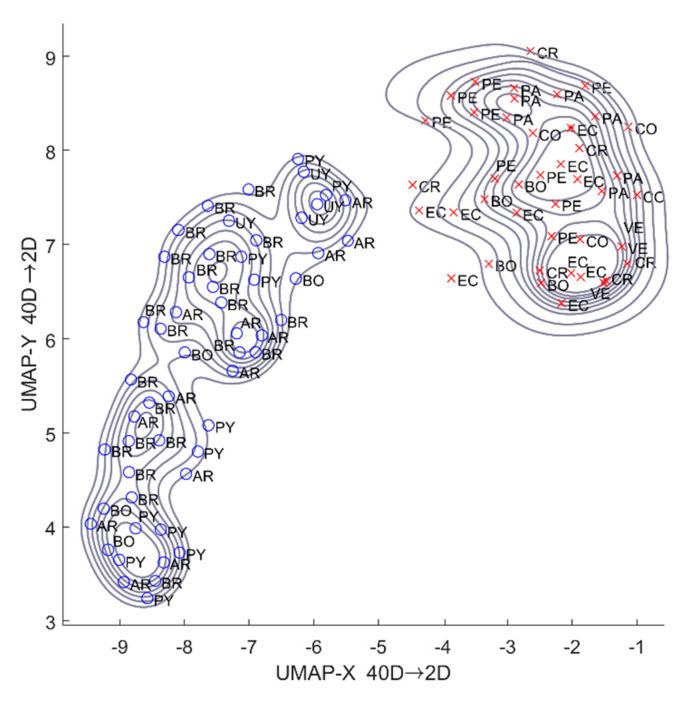
UMAP visualization and k-medoids clustering of the *N* = 101 samples of Rough-legged Tyrannulet (*sensu lato*) included in the study. Two clusters are identified using markers: Group A with blue circles, Group B with red x-mark. Each individual is labeled with the country of precedence using country codes alpha-2.

**Figure 4 animals-10-01406-f004:**
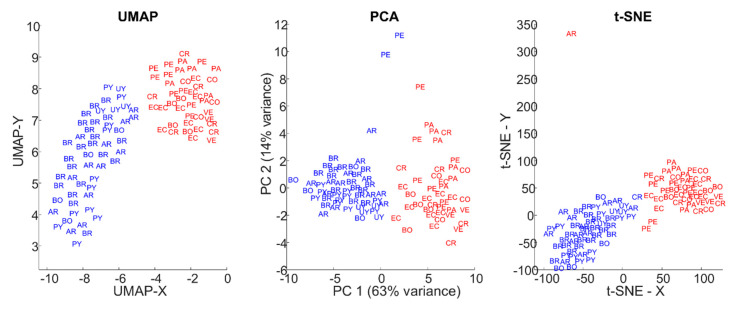
Comparison of results using UMAP, principal components analysis (PCA), and t-distributed stochastic neighbor embedding (t-SNE). Each individual is labeled with the country of precedence using country codes alpha-2, and groups are marked using text color: Group A in blue, Group B in red.

**Figure 5 animals-10-01406-f005:**
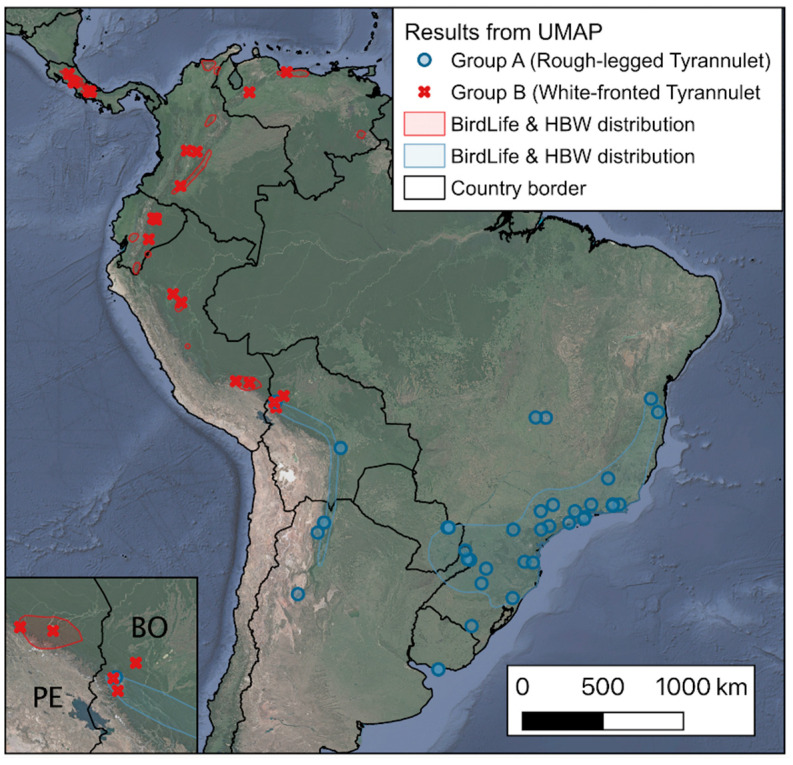
Segregation by the distribution of two different acoustic groups within the Rough-legged Tyrannulet complex.

**Figure 6 animals-10-01406-f006:**
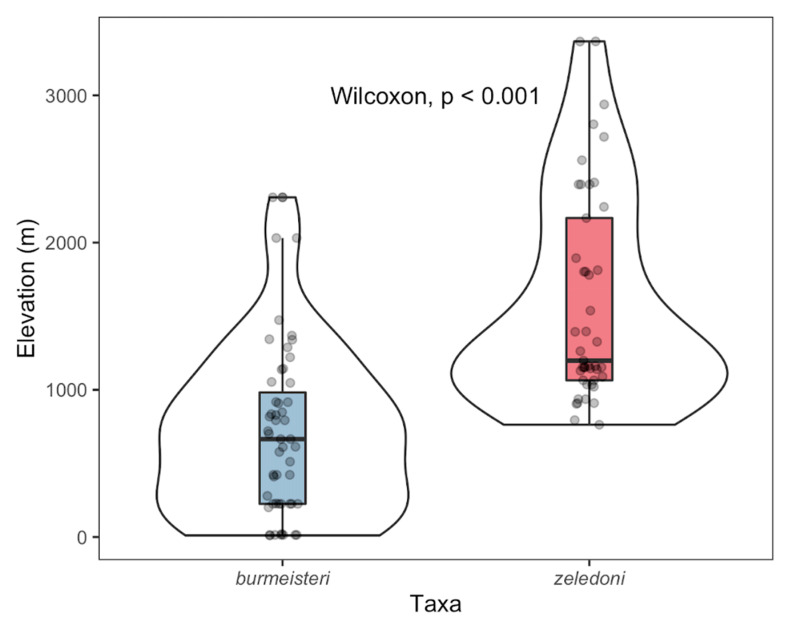
Elevation comparison between two groups of Rough-legged Tyrannulet (*sensu lato*). Rough-legged Tyrannulet (*sensu strictu*; *Phyllomyias burmeisteri*) and White-fronted Tyrannulet (*P. zeledoni*) differed in median elevation. White violin plots show the density of records in the elevation axis. Box plots show the interquartile range (IQR) from the first and third quartile (25 and 75% data), while whiskers extend up to ±1.5 × IQR.

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
