# Peer review of "Uniform Manifold Approximation and Projection for Clustering Taxa through Vocalizations in a Neotropical Passerine (Rough-Legged Tyrannulet, Phyllomyias burmeisteri)"

_animals, 2020, doi:10.3390/ani10081406_

Round 1

Reviewer 1 Report

The submitted paper evaluates and attempts to showcase the Uniform Manifold Approximation and Projection (UMAP) as a unique and advantageous method for separation of population groups in the vocalization analysis of Tyrannulet bird.

Manuscript would significantly benefit by a native English proofreader as there are many grammar and language errors. The coherence and logical flow/sequence is lacking in the manuscript.

The paper should be rearranged to provide a clear flow of the content. One of the concerns in this study is the usage of limited dataset (only 10 notes from 46 birds - 46 individuals and 40 dimensions is pretty low). Information concerning the uniform distribution and the local constant detail of Riemannian manifold is lacking. 

I suggest the authors to resubmit the manuscript for consideration by carrying out analysis using a large data set and using tSNE and another method such as PCA or LDA for comparative analysis.

Line 38 - 'Specimens' or 'Species'

Line 44 - Justification is not very clear. PCA always provides dimensionality reduction with a certain amount of overlap.

Lines 134 to 155 - These are basically the theory of UMAP and how the dimensionality reduction is being carried out. These are commonly available information and does not add any specific information or value to this work.

Section 4. Discussion

Cross-validation is missing. Have you run the dimensionality reduction with other comparable methods to provide information on the unique advantages of UMAP in the grouping of species taxonomy?

Without cross validation and large data set, the provided data set and analysis is not convincing.

Reviewer 2 Report

My comments for the authors appear in the attached file. 

Round 2

Reviewer 1 Report

The authors appeared to have carried out suggested changes through the review process. 

Reviewer 2 Report

The authors have made extensive improvements to the manuscript, which I recommend for publication. I downloaded the supplementary audio files but was not able to open them on my computer, which did not recognize the format. In order to be a useful addition to this paper, the audio files should be in a standard format, such as mp3 or wav.